# Aicardi–Goutières Syndrome with Congenital Glaucoma Caused by Novel *TREX1* Mutation

**DOI:** 10.3390/jpm13111609

**Published:** 2023-11-15

**Authors:** Marta Świerczyńska, Agnieszka Tronina, Erita Filipek

**Affiliations:** 1Department of Ophthalmology, Faculty of Medical Sciences in Katowice, Medical University of Silesia, 40-514 Katowice, Poland; 2Department of Ophthalmology, Kornel Gibiński University Clinical Center, Medical University of Silesia, 40-514 Katowice, Poland; 3Department of Pediatric Ophthalmology, Faculty of Medical Sciences in Katowice, Medical University of Silesia, 40-514 Katowice, Poland; agnieszka.tronina@sum.edu.pl (A.T.); erita.filipek@sum.edu.pl (E.F.); 4Department of Pediatric Ophthalmology, Kornel Gibiński University Clinical Center, Medical University of Silesia, 40-514 Katowice, Poland

**Keywords:** Aicardi–Goutières syndrome, *TREX1*, congenital glaucoma, interferon alpha, type 1 interferonopathies

## Abstract

Background: Aicardi–Goutières syndrome (AGS) is a rare genetic disorder characterized by microcephaly, white matter lesions, numerous intracranial calcifications, chilblain skin lesions and high levels of interferon-α (IFN-α) in the cerebrospinal fluid (CSF). However, ocular involvement is reported significantly less frequently. Case presentation: We present a case of a neonate with hypotrophy, microcephaly, frostbite-like skin lesions, thrombocytopenia, elevated liver enzymes and hepatosplenomegaly. Magnetic resonance imaging (MRI) of the brain showed multiple foci of calcification, white matter changes, cerebral atrophy, and atrophic dilatation of the ventricular system. The inflammatory parameters were not elevated, and the infectious etiology was excluded. Instead, elevated levels of IFN-α in the serum were detected. Based on the related clinical symptoms, imaging and test findings, the diagnosis of AGS was suspected. Genetic testing revealed two pathogenic mutations, c.490C>T and c.222del (novel mutation), in the three prime repair exonuclease 1 (*TREX1*) gene, confirming AGS type 1 (AGS1). An ophthalmologic examination of the child at 10 months of age revealed an impaired pupillary response to light, a corneal haze with Haab lines in the right eye (RE), pale optic nerve discs and neuropathy in both eyes (OU). The intraocular pressure (IOP) was 51 mmHg in the RE and 49 in the left eye (LE). The flash visual evoked potential (FVEP) showed prolonged P2 latencies of up to 125% in the LE and reduced amplitudes of up to approximately 10% OU. This girl was diagnosed with congenital glaucoma, and it was managed with a trabeculectomy with a basal iridectomy of OU, resulting in a reduction and stabilization in the IOP to 12 mmHg in the RE and 10 mmHg in the LE without any hypotensive eyedrops. Conclusions: We present the clinical characteristics, electrophysiological and imaging findings, as well as the genetic test results of a patient with AGS1. Our case contributes to the extended ophthalmic involvement of the pathogenic c.490C>T and c.222del mutations in *TREX1.*

## 1. Introduction

Aicardi–Goutières syndrome (AGS) is a rare autoimmune neurological disorder belonging to type I interferonopathies. There are seven subtypes based on the different pathogenic genes: three prime repair exonuclease 1 (*TREX1)* (AGS1), *RNASEH2B* (AGS2), *RNASEH2C* (AGS3), *RNASEH2A* (AGS4), *SAMHD1* (AGS5), *ADAR1* (AGS6) and *IFIH1* (AGS7) [1,2]. Mutations affecting *RNASEH2B* and *TREX1* are reported to be the most common, representing 35% and 17% of all cases, respectively [3]. AGS is usually inherited in an autosomal recessive manner. However, there may also be de novo or inherited autosomal dominant pathogenic variants in *TREX1* or *ADAR*, as well as heterozygous autosomal dominant pathogenic variants in *IFIH1*. Mutations in the above genes affect the targeting and/or metabolism of nucleic acids, thereby promoting a type I interferon (IFN-I)-mediated innate immune response [1,2,3,4,5,6].

The major clinical features of AGS include encephalopathy, significant intellectual disability, acquired microcephaly during the first year of life, dystonia, spasticity, sterile pyrexias, intracranial calcifications, white matter lesions, brain atrophy, bilateral striatal necrosis, chilblain lesions on the feet, hands, ears or more diffuse throughout the skin [1,2,7,8,9]. The characteristic features include lymphocytosis, high levels of interferon α (IFN-α) in the cerebrospinal fluid (CSF) and serum with an increased expression of interferon-stimulated genes (ISGs) in the peripheral blood—the so-called “interferon signature” [1,2,6,10]. Moreover, patients with AGS may demonstrate intracerebral vasculopathy, hepatosplenomegaly, elevated liver enzymes, thrombocytopenia, hemolytic anemia, elevated autoantibodies, hypothyroidism, insulin-dependent diabetes mellitus, transitory antidiuretic hormone deficiency, neonatal cardiomyopathy and demyelinating peripheral neuropathy [7,9,11,12]. 

With new sequencing technologies becoming more extensively used in routine clinical practice, the spectrum of symptoms coexisting with AGS continues to expand. However, the amount of data available regarding the ocular complications associated with AGS is still quite limited. We present the clinical characteristics, electrophysiological and imaging outcomes as well as genetic test results of a patient with AGS1. Our case provides a more comprehensive insight into the affinity of pathogenic c.490C>T and c.222del (novel variant) mutations in *TREX1* for ocular symptoms and complications.

## 2. Case Report

We present the case of a Caucasian girl, the first child of a healthy couple without consanguinity, with a family history of autoimmune and genetic diseases with no detectable pathologies. The girl was born at 37 weeks’ gestation with an Apgar score of 7/8/8/9; the birth weight was 2370 g (<10th centile, hypotrophy), and the head circumference was 29.8 cm (<3rd centile, microcephaly). A cranial ultrasound performed on the first day of life revealed dilated vascular plexuses with irregular contours, a dilated ventricular system, leukomalacia involving the frontal horns and indistinct echogenicity of the brain tissue. Subsequently, magnetic resonance imaging (MRI) of the brain was performed, demonstrating generalized cortical and subcortical atrophy of both cerebral hemispheres, focal areas of cerebral malacia in the frontal, parietal, temporal and occipital lobes, together with an atrophic dilated ventricular system. Numerous small calcifications and hemorrhagic foci were found, most prominently in the white matter of the periventricular structures and deep cerebral hemispheres. Furthermore, thinning of the corpus callosum and pontocerebellar hypoplasia were found.

Initially, a congenital infection and a pre-existing encephalitis were suspected, but the inflammatory parameters on the first day of life were negative, and the polymerase chain reaction (PCR) test from the blood and cerebrospinal fluid (CSF) excluded infection. However, the enzyme-linked immunosorbent assay (ELISA) showed that the serum IFN-α and IFN-β levels were both abnormally elevated. The screening for inborn metabolic disorders was negative. The newborn also had moderate thrombocytopenia, elevated liver enzymes and hepatosplenomegaly. Moreover, chilblain-like lesions on the fingers and toes and scattered petechiae were present. In the following weeks, no growth in the circumference of the head and increased muscle tone in the four limbs were noted. A repeated brain MRI scan performed after two months showed the evolution of hypoxic-ischemic changes progressing towards encephalomalacia and atrophy.

The diagnosis of AGS was suspected based on the concomitance of clinical symptoms, imaging, and diagnostic test results. Whole-exome genome sequencing (WES) was performed, revealing the presence of two mutations in the *TREX1* gene. The first mutation, c.490C>T p.(Arg164Ter), is registered as a pathogenic variant affecting the function of the protein encoded by *TREX1*, and it is associated with the clinical manifestations of AGS. However, the second mutation, c.222del p.(Lys75ArgfsTer13), was not previously registered in the databases and is a frameshift mutation resulting in the loss of function of *TREX1*. The father presented a c.490C>T p.(Arg164Ter), and the mother presented a c.222del p.(Lys75ArgfsTer13). The parents of the proband were heterogeneous carriers with normal phenotypes, while the child was diagnosed with AGS1.

This patient had an ophthalmological examination on the first day of life. At that time, it was assessed that the intraocular pressure (IOP) was normal, the anterior and posterior eye segment examination was without abnormalities and the retina revealed no inflammatory changes in both eyes (OU). At 10 months of age, the girl was referred to the Ophthalmology Outpatient Clinic for a follow-up examination, which showed an impaired pupillary response to light OU. An ophthalmic examination performed under general anesthesia showed slight corneal haze with Haab lines in the RE. The corneal diameter was 11.5 mm × 11.5 mm in the RE and 11 × 11.5 in the LE. A gonioscopy revealed a moderately deep, hypoplastic (greater in the RE) iridocorneal angle, a hypoplastic iris with prominent vessels and a poorly defined ciliary body, without pigment OU. The IOP measured with the iCare tonometer was 51 mmHg in the RE and 49 mmHg in the LE. The corneal pachymetry was, on average, 707 μm in the RE and 712 μm in the LE. A fundus examination revealed pale optic nerve discs with advanced glaucomatous neuropathy (Figure 1), macula with no reflex and a poorly pigmented retinal OU. Additionally, in the flash visual evoked potential (FVEP) performed, the P2 latencies were normal in the RE and prolonged to 125% in the LE, and the amplitudes were reduced to approximately 10% OU.

According to these findings, the girl was diagnosed with congenital glaucoma and subsequently underwent trabeculectomy with basal iridectomy OU (Figure 2A–H).

After one month, the IOP was 12 mmHg in the RE and 10 mmHg in the LE without any hypotensive agents. The girl remained under the constant care of the Ophthalmology Outpatient Clinic, with the IOP maintained within normal limits 2 years after the surgery (Figure 3). As an additional feature, the presence of slight lagophthalmos during the patient’s sleep was observed, resulting in exposure keratopathy.

## 3. Discussion

We can distinguish the two clinical presentations of AGS. The early-onset form, which starts in utero, features a severe course after birth and resembles a congenital infection (pseudo-TORCH) with a high risk of death. However, the late-onset form, in which an initially healthy infant presents with a subacute onset of profound neurological regression only after a few months, has a greater variety of symptoms [7,12,13]. Neonatal presentation is most often associated with a mutation in the *TREX1*, as in the case of the patient we described. The *TREX1* gene is located on chromosome 3p21 and encodes 3′ -> 5′ endonuclease involved in degrading single-stranded (ss) and double-stranded (ds) DNA substrates in vitro [14,15]. The loss-of-function mutations in *TREX1* results in an abnormal accumulation of nucleotides in the cytoplasm, which can be bound by the DNA sensor—the cyclic GMP-AMP synthase (cGAS). This is followed by the activation of the IFN signaling pathway through the stimulator IFN gene (STING), leading to an increased expression of IFN-I and an enhanced immune response [16]. This response is analogous at the cellular level to that produced by an exposure to viral nucleic acids (DNA and RNA), which explains why the AGS has a phenotype similar to intrauterine viral infection. Unfortunately, many cases of AGS remain undiagnosed, and an accurate diagnosis usually occurs late and is commonly associated with the birth of a second affected child. For this reason, AGS should be considered in neonates with features of congenital infection in which the pathogen has not been isolated [17] (Table 1).

Glaucoma has been found to be the most common concomitant ophthalmological condition with AGS, with the majority of the symptoms beginning in the first months of life [1,2,7,9,18,19]. However, there has also been a case reported of a patient with AGS in whom bilateral glaucoma requiring treatment appeared at the age of 6 [2]. According to the major report on individuals with AGS, glaucoma was diagnosed in 6.3%, of which 20.8% were patients with a pathological *SAMHD1* variant. However, among patients with abnormal *ADAR* and *IFIH1* variants, glaucoma was not detected [2]. Various other ophthalmic symptoms have been described so far: conjunctivitis, dry eye, exposure keratopathy, corneal perforation, posterior synechiaea, aniridia, spherophakia, optic atrophy, papillitis, optic neuritis, cortical blindness, choroidal thickening, nystagmus and nanophthalmos [1,9,20,21,22]. In addition, brain white matter abnormalities with prolonged latencies on VEP testing were detected in some children, indicating a degree of cortical blindness, while the electroretinography (ERG) photopic responses performed so far were normal [22,23].

The other example of an immunogenetic disorder associated with the induction of IFN-I production and the increased expression of ISGs in which glaucoma occurs is Sigleton Merten syndrome (SGMRT). It is an autosomal-dominant condition caused by gain-of-function variants in RIG-I-like receptor proteins. The normal function of the receptor is to recognize exogenous dsRNA, thereby activating innate immune pathways and IFN-I signaling as part of the antiviral response. The systemic features involve a psoriasiform rash, vascular calcifications, skeletal dysplasia and dental anomalies, whereas the ocular manifestations include congenital or juvenile open-angle glaucoma (OAG) [21,24,25].

In the literature, several cases have been reported in which the administration of IFN resulted in the development or progression of glaucoma (at the same time, the use of ribavirin or protease inhibitors has not been directly linked to an increase in the IOP) [26,27,28]. Ilyas et al. described the case of a 51-year-old woman with primary OAG (POAG) with an increase in the IOP one month after starting triple therapy treatment (consisting of pegIFN-α-2a, ribavirin and boceprevir) for chronic hepatitis C. Although treated with hypotensive drops, the IOP remained above average and declined significantly only after the discontinuation of antiviral therapy [26]. A case of a man with hepatitis C was also described in whom the use of pegIFN-α-2b and ribavirin induced an exacerbation of already diagnosed glaucoma [27]. However, Kwon et al. reported the development of POAG in a 15-year-old boy with chronic hepatitis B during treatment with IFN-α, which disappeared with the cessation of the IFN therapy [28]. The mechanism by which IFN therapy may lead to an increase in the IOP remains unclear. It has been shown that IFN, through a stimulation of the effector function of immune cells, exhibits immunomodulatory properties [29]. Thus, it has been suggested that increased levels of IFN-α may be associated with the activation of leukocytes (monocytes, macrophages and dendritic cells). Moreover, the secretion of macrophage migration inhibitory factor (MIF) by the human trabecular meshwork (TM) increases the expression of IFN and thus enhances the T helper cytokines and contributes to an imbalance between different cytokines and cells. TM is the site of the highest resistance to aqueous humor outflow and is thought to be critical to the regulation of the IOP. One of the important components in the resistance to outflow through the TM is the extracellular matrix (ECM). The MIF induces an upregulation of matrix metalloproteinases-1 and -3, in a dose-dependent way, and may induce ECM degradation [30]. Additionally, it was demonstrated that there were significant differences in the cytokine profile (including increased IFN-α concentration) in the aqueous humor obtained from the anterior chamber of glaucomatous eyes compared to the control group. In addition, it was noted that in POAG, an aqueous inflammatory response was more extensive compared to patients with primary angle closure glaucoma [31].

The continuous development of obstetric ultrasonography provides an opportunity to monitor the development of the eyeballs as well as measure their axial lengths earlier and more accurately, especially after the 17th gestational week [32]. MRI is another, although not yet so accessible, method used to diagnose congenital defects. To date, no negative effects of this examination on the fetus have been demonstrated, but it is discouraged during the first trimester [33]. Identifying megalophthalmos, suggesting congenital glaucoma in the fetus, makes it possible to plan and perform glaucoma surgery in the newborn as early as possible, potentially increasing the chances of maintaining the development of normal vision [34]. In addition, extensive research is being conducted to identify biomarkers that allow for the prenatal detection of congenital glaucoma. To date, cytochrome p4501B1 deficiency has been shown to affect abnormal ocular tissue during early fetal development, and fetuses with cytochrome p4501B1 mutations are more likely to develop congenital glaucoma [35].

There is currently no specific and effective treatment for AGS. Conventional immunosuppressive drugs (intravenous methylprednisolone or immunoglobulins) have not been successful [36,37]. New therapeutic strategies focused on the reduction in INF-I production and/or the inhibition of the IFN-I-induced signaling pathway. Among the investigated drugs, Janus kinase (JAK) inhibitors (baracitinib, ruxolitinib and tofacitinib) especially demonstrate promising effects [13,25]. Their abilities to cause a reduction in neurological symptoms, improve neuro-motor skills as well as cause beneficial effects on proinflammatory biomarkers in the serum and CSF have been proven [38]. Likewise, the use of reverse transcriptase inhibitors (RTIs) has been shown to reduce the INF-α levels [39]. Anti-IFN-α antibodies (sidalimumab), anti-IFN-I receptor antibodies (anifrolumab) or molecules inhibiting the cGAS-STING pathway (suramin) may also be useful [13,25,40]. It remains unclear, though, whether the above immunomodulatory therapy may be useful in the prevention or treatment of AGS-associated glaucoma.

## 4. Conclusions

Given the similar presentation to viral infection, AGS should be considered during the differential diagnosis in neonates with a phenotype of congenital infection. It is important from clinical genetics perspective when considering recurrence risks. Congenital glaucoma can be considered as a part of the phenotypic spectrum of AGS; therefore, regular ophthalmological examinations are essential, especially in the first years of a child’s life, in order to diagnose the disease and promptly initiate treatment.

## Figures and Tables

**Figure 1 jpm-13-01609-f001:**
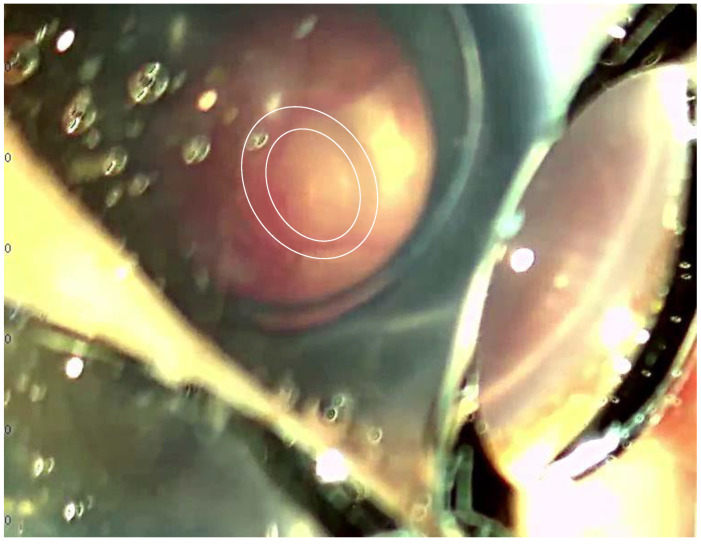
Photography of the RE fundus, obtained during examination under general anesthesia, showed an enlarged cup-to-disc ratio, nasal shifting of blood vessels and pallor of the remaining neuro-retinal rim.

**Figure 2 jpm-13-01609-f002:**
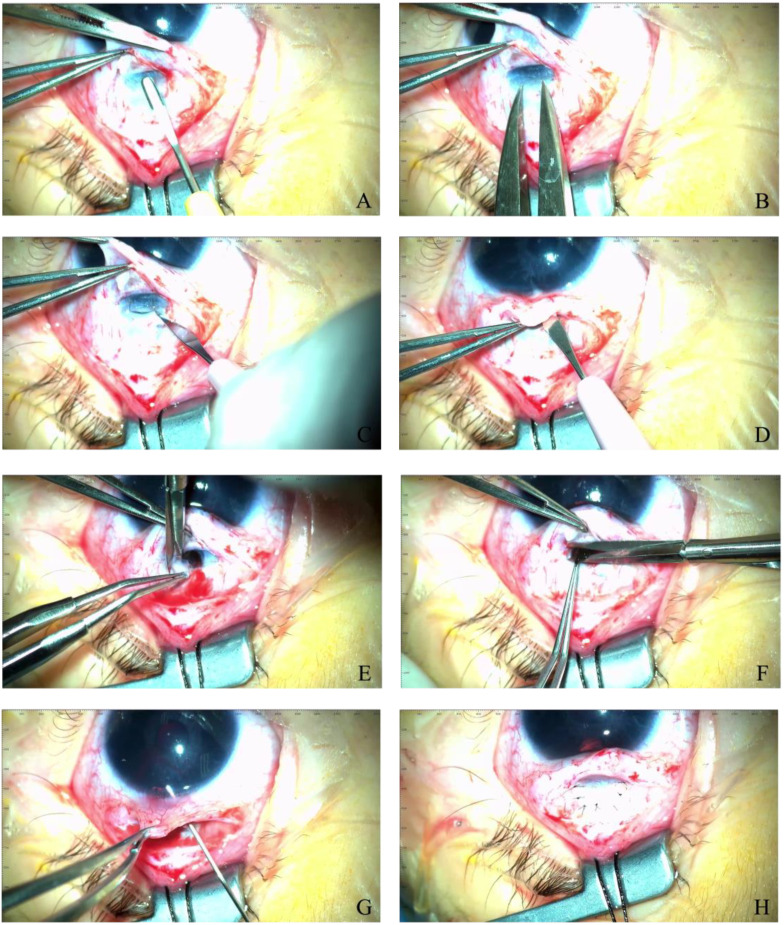
Trabeculectomy with basal iridectomy in the RE: (**A**) preparation of the superficial scleral flap (limbal-based flap, 4 mm × 3 mm); (**B**) determination of the size of the deep scleral flap; (**C**) dissection of the deep scleral flap (1.5 mm × 2 mm); (**D**) punctate opening of the anterior chamber; (**E**) excision of the deep scleral flap; (**F**) basal iris excision; (**G**) inspection of the created fistula; (**H**) superficial flap suture placement.

**Figure 3 jpm-13-01609-f003:**
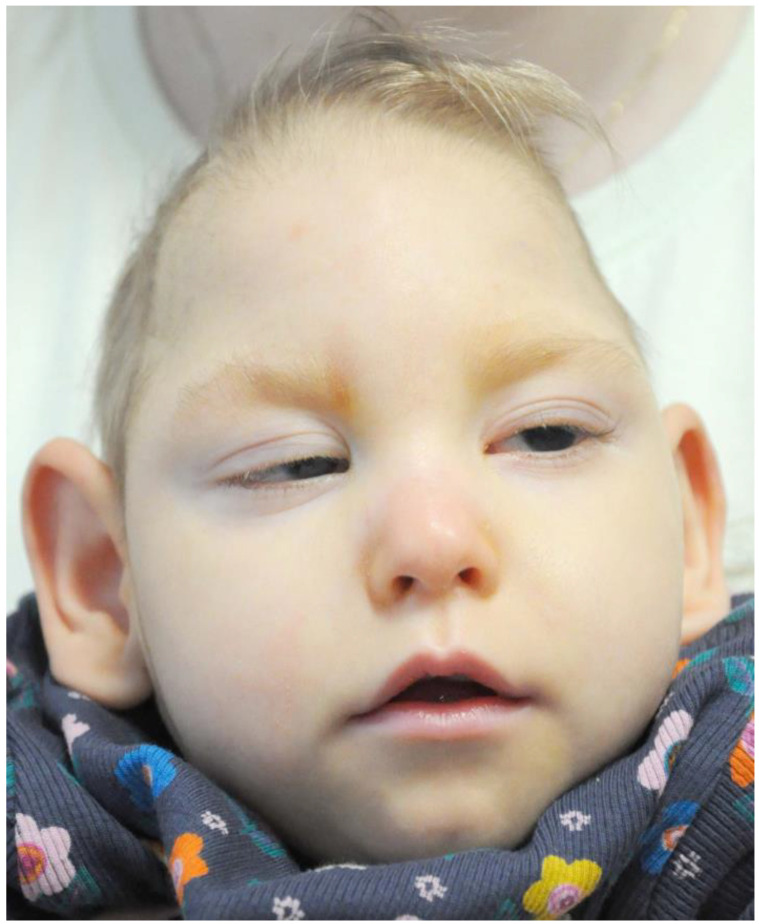
Photography of a patient at the age of 3 years. There is apparent small neurocranium, oval face and fairly large auricles; forehead is slightly posteriorly tilted; fine, short hair with V-shaped line on head; mouth with narrow red lips.

**Table 1 jpm-13-01609-t001:** *TREX1* pathogenic variants according to the Human Gene Mutation Database (HGMD).

**Missense/Nonsense Mutation**			
**HGMD Codon Change**	**HMD Amino Acid Change**	**HGVS (Nucleotide)**	**HGVS (Protein)**
ACC-AAC	Thr13Asn	c.38C>A	p.T13N
ACC-CCC	Thr13Pro	c.37A>C	p.T13P
ATC-ATG	Ile15Met	c.45C>G	p.I15M
GAC-CAC	Asp18His	c.52G>C	p.D18H
ACG-AGG	Thr32Arg	c.95C>G	p.T32R
TGT-TGA	Cys42Term	c.126T>A	p.C42
CCG-CAG	Pro61Gln	c.182C>A	p.P61Q
AAG-AGG	Lys66Arg	c.197A>G	p.K66R
CTG-CCG	Leu69Pro	c.206T>C	p.L69P
CTG-CAG	Leu92Gln	c.275T>A	p.L92Q
CGT-CAT	Arg97His	c.290G>A	p.R97H
CGC-CAC	Arg114His	c.341G>A	p.R114H
GTG-GCG	Val122Ala	c.365T>C	p.V122A
CTG-CCG	Leu162Pro	c.485T>C	p.L162P
CGA-TGA	Arg164Term	c.490C>T	p.R164
CAC-TAC	His195Tyr	c.583C>T	p.H195Y
GAG-AAG	Glu198Lys	c.592G>A	p.E198K
GAT-AAT	Asp200Asn	c.598G>A	p.D200N
GAT-CAT	Asp200His	c.598G>C	p.D200H
GTC-GAC	Val201Asp	c.602T>A	p.V201D
TGG-TAG	Trp210Term	c.629G>A	p.W210
GCC-ACC	Ala223Thr	c.667G>A	p.A223T
ACA-CCA	Thr303Pro	c.907A>C	p.T303P
**Small deletions**			
**HGMD deletion**		**HGVS (nucleotide)**	**HGVS (protein)**
CCCCC^**^49^**ACCTCtcAGGGGCCACC		c.150_151delTC	p.(Gln51Glyfs*50)
CCCACC^**^50^**TCTCagGGGCCACCTC		c.152_153delAG	p.(Gln51Argfs*50)
CCTGC^**^78^**AGCCCtgcagccAGCGAGATCA		c.237_243delTGCAGCC	p.(Ala81Argfs*5)
GCCCT^**^80^**GCAGCcagcGAGATCACAG		c.243_246delCAGC	p.(Ser82Argfs*5)
TACGAC^**^131^**TTCCcCCTGCTCCAA		c.397delC	p.(Leu133Cysfs*27)
GTGGAT^**^155^**AGCAtCACTGCGCTG		c.467delT	p.(Ile156Thrfs*4)
GATAGC^**^156^**ATCAcTGCGCTGAAG		c.470delC	p.(Thr157Metfs*3)
CGAGCA^**^166^**AGCAgCCCCTCAGAA		c.500delG	p.(Ser167Thrfs*13)
AGGAAG^**^176^**AGCTaTAGCCTAGGC		c.530delA	p.(Tyr177Leufs*3)
GCTCAGC^**^207^**ATCtgtcaGTGGAGACCA		c.622_626delTGTCA	p.(Cys208Valfs*31)
AGCCA^**^244^**AGACCaTCTGCTGTCA		c.735delA	p.(Ser246Leufs*31)
AAGGAC^**^279^**CCTGgAGCCCTATCC		c.839delG	p.(Gly280Glufs*18)
CCAGG^**^285^**GAGGGgCTGCTGGCCC		c.858delG	p.(Leu287Cysfs*11)
GCTGCTG^**^289^**GCCccactgggtctgctggccATCCTGACCT		c.868_885del18	p.(Pro290_Ala295del)
**Small insertions**			
**HGMD insertion**		**HGVS (nucleotide)**	**HGVS (protein)**
TTTCGAC^**^19^**ATGgGAGGCCACTG		c.58dupG	p.(Glu20Glyfs*82)
GAGCCCC^**^48^**CCCcACCTCTCAGG		c.144dupC	p.(Thr49Hisfs*53)
CCTGTGT^**^71^**GTGtgGCTCCGGGGA		c.212_213dupTG	p.(Ala72Trpfs*17)
CAGCCCT^**^80^**GCAaGCCAGCGAGA		c.240dupA	p.(Ala81Serfs*21)
CCCTGCA^**^81^**GCCctgcagccAGCGAGATCA		c.236_243dupCTGCAGCC	p.(Ser82Leufs*9)
ACAGGT^**^87^**CTGAagGCACAGCTGT		c.262_263insAG	p.(Ser88Lysfs*23)
TGGGCGT^**^98^**CAAaTGTTTTGATG		c.294dupA	p.(Cys99Metfs*3)
GTCAA^**^99^**TGTTTgtttTGATGACAAC		c.296_299dupGTTT	p.(Phe100Leufs*3)
GCCTG^**^122^**GTGGCggcACACAATGGT		c.366_368dupGGC	p.(Ala123dup)
GCTCCAA^**^136^**GCAccccctgctccaagcaGAGCTGGCTA		c.393_408dup16	p.(Glu137Profs*24)
TGAGGGT^**^200^**GATgatGTCCTGGCCC		c.599_601dupATG	p.(Asp200dup)
ATCTGT^**^209^**CAGTcagtGGAGACCACA		c.625_628dupCAGT	p.(Trp210Serfs*32)
TGTCAG^**^210^**TGGAtggaGACCACAGGC		c.628_631dupTGGA	p.(Arg211Metfs*31)
CATCAGG^**^231^**CCCcATGTATGGGG		c.693dupC	p.(Met232Hisfs*9)
**Gross insertions**			
**DNA level**		**Insertion/duplication**	**Description**
gDNA		duplication	54 bp c.609_662

HGVS—Human Genome Variation Society.

## Data Availability

Data are contained within the article.

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
