# Peer review of "Aicardi–Goutières Syndrome with Congenital Glaucoma Caused by Novel TREX1 Mutation"

_jpm, 2023, doi:10.3390/jpm13111609_

Round 1

Reviewer 1 Report

Comments and Suggestions for Authors

1- Is there a consent letter from the patient?

2- What is the approval number for the ethical committee?

3- In line 198 of the discussion section, how could immunomodulation lead to molecular changes in the trabecular meshwork?

4- Could you explain how to diagnose congenital glaucoma in a fetus during the third trimester of pregnancy, and if there are any treatment options during this period?

5- The references need to be updated.

Comments on the Quality of English Language

1- The grammar and spelling need to be revised.

Author Response

Thank you for your time devoted to reviewing our manuscript.

We are grateful for all your comments.

We have highlighted changes made to the main text in the red color.

1- Is there a consent letter from the patient?

Yes, we obtained informed written consent from the patient's guardians. This was sent to the editor during manuscript submission.

2- What is the approval number for the ethical committee?

This article is a case report and has a retrospective character. All the investigations performed, treatment and interventions applied were essential and the child's guardians provided their informed consent.

3- In line 198 of the discussion section, how could immunomodulation lead to molecular changes in the trabecular meshwork?

A section outlining this issue more precisely has been added (lines 213-220).

4- Could you explain how to diagnose congenital glaucoma in a fetus during the third trimester of pregnancy, and if there are any treatment options during this period?

Thank you very much for this specific question. We included a section in the discussion describing the topic more extensively (lines 225-237).

5- The references need to be updated.

References have been updated.

1- The grammar and spelling need to be revised.

Language proofreading of the text was conducted.

Reviewer 2 Report

Comments and Suggestions for Authors

This is an interesting to read case report about a patient with Aicardi-Goutieres syndrome (AGS) with a novel TREX1 mutation. AGS is a very rare immunogenetic disorder with mutations in 7 known genes and was first described in 1984. There are several systemic and ocular manifestations described including a higher risk for glaucoma – as it is the case in this patient, first diagnosed 10 months after birth. A trabeculectomy was necessary to control the high intraocular pressure in both eyes with advanced glaucomatous changes of the optic disc. Besides the clinical characteristics of this patient genetic, electrophysiological, and imaging results were demonstrated. In summary: an informative and excellent written case report.

Comments:

Figure 2. shows the detailed procedures in trabeculectomy and iridectomy surgery. These surgical steps are well known. Wouldn´t it be advantageous to show some of the MR-images?

Did the authors use mitomycin C to avoid early scaring of the bleb?

Do the authors have an explanation why the latencies of the VEP of the right eye was normal, but those of the left eye prolonged to 125%?

Figure 3. shows the patient en face. The authors describe parts of her face: “… Eyelid fissures in slightly oblique-upper position. Relatively prominent, broad, well pointed nose at the base. Mouth with narrow red lips with the corners turned downwards…” My impression is that the eye lids are in no oblique position, that the base of the nose seems to be not prominent or broad, and that the corners of the lips don´t turn downwards. But I might wrong…

Maybe the authors might mention in their excellent discussion, that SAMHD1 mutation was found to be most associated with glaucoma with one study showing that over 20% of patients with this mutation have congenital glaucoma (Ref 21.). The ADAR and IFIH1 mutations were found to be the least associated with glaucoma.

Line 113: should be iCare (not ICare), 168: should be posterior

Author Response

Thank you for your time devoted to reviewing our manuscript.

We are grateful for all your comments.

We have highlighted changes made to the main text in the red color.

Figure 2. shows the detailed procedures in trabeculectomy and iridectomy surgery. These surgical steps are well known. Wouldn´t it be advantageous to show some of the MR-images?

We absolutely agree with the suggestion that the addition of an MRI would be a beneficial thing. Unfortunately, this examination was performed in another facility, and we are unable to publish these scans.

Did the authors use mitomycin C to avoid early scaring of the bleb?

Mitomycin C was not used in the described procedures. It is used in our clinical center in the case of children with severe goniodysgenetic changes, in aphakic eyes, with glaucoma in the course of uveitis, after prior glaucoma surgery or when there are concomitant diseases associated with an increased risk of scarring.

Do the authors have an explanation why the latencies of the VEP of the right eye was normal, but those of the left eye prolonged to 125%? 

We cannot specifically identify the cause of this result. It could be due to either the perinatal hypoxia or an impaired myelination.

Figure 3. shows the patient en face. The authors describe parts of her face: “… Eyelid fissures in slightly oblique-upper position. Relatively prominent, broad, well pointed nose at the base. Mouth with narrow red lips with the corners turned downwards…” My impression is that the eye lids are in no oblique position, that the base of the nose seems to be not prominent or broad, and that the corners of the lips don´t turn downwards. But I might wrong…

Thank you for this feedback. The grimace on the child's face captured in the photograph may indeed not fully reflect the dimorphic features which we observed when investigating the patient in real time. We removed the questionable sections to avoid misleading the readers.

Maybe the authors might mention in their excellent discussion, that SAMHD1 mutation was found to be most associated with glaucoma with one study showing that over 20% of patients with this mutation have congenital glaucoma (Ref 21.). The ADAR and IFIH1 mutations were found to be the least associated with glaucoma.

Thank you very much for this suggestion. We introduced modifications to the text.

Line 113: should be iCare (not ICare), 168: should be posterior

Thank you for pointing out these mistakes, this has been corrected.

Reviewer 3 Report

Comments and Suggestions for Authors

Swierczyñska, M., et al have elucidated the clinical manifestations, electrophysiological and radiological findings, alongside genetic test outcomes, in the case of an individual afflicted with Aicardi-Goutières syndrome (AGS), an exceedingly rare genetic ailment distinguished by microcephaly, white matter lesions, multiple intracranial calcifications, chilblain skin lesions, etc.

The authors serve to augment the existing understanding of the ocular involvement associated with the pathogenic mutations c.490C>T and c.222del within the TREX1 gene. The authors provide insights into the current clinical characteristics, electrophysiological and imaging outcomes as well as genetic test results of a patient with AGS1. According to the authors, the novelty of this research can help improve the quantity of the amount of available data on the ocular complications associated with AGS, as well as contribute toward the expansion of the ophthalmological involvement of pathogenic mutations in the TREX1 gene.

However, the concerning part of this study is that all the results are based on a single patient (mentioned by the authors). This study highlights the case of a Caucasian girl born with microcephaly and various brain abnormalities, initially suspected of congenital infection. However, further tests revealed elevated interferon levels, thrombocytopenia, liver issues, and chilblain-like skin lesions, leading to a diagnosis of Aicardi-Goutières syndrome (AGS). Genetic testing identified two mutations in the TREX1 gene, one known to be pathogenic and the other a novel frameshift mutation. The child's parents were carriers of these mutations but had no symptoms. The main takeaway message is that AGS can mimic viral infections in neonates, and should be considered in the differential diagnosis with regular ophthalmological examinations for early AGS detection, aiding in effective management and considering recurrence risks in clinical genetics.

Nonetheless, the article seemed to possess no major concerns, as it is a straightforward study of mutation, c.222del p.(Lys75ArgfsTer13), in the TREX1 gene associated with AGS1. Overall, the clarity of the text is good and easily understandable but a few of the points need to be discussed properly with references and especially a table with all TREX1 gene mutations previously reported. The consent from the parent is a must (as provided) which is highly appreciated.

The manuscript has very few typographical and grammatical errors which need to be corrected. In general, the manuscript can accomplish the caliber of quality for consideration for publication in the Journal “Personalized Medicine”. The authors are advised to consider the comments below:

Comments

1.      Figure 1 / The image is not clear to understand. Please mark the pale optic disk and the marking of the advanced glaucomatous neuropathy.

2.      Please include a supplementary table with all TREX1 gene mutations previously reported for AGS.

3.      Page 4/ Line 137 / “The child developed exposure keratopathy OU due to slight lagophthalmos during sleep”  - Could you please explain what are you trying to establish with this statement?

Comments on the Quality of English Language

The manuscript has very few typographical and grammatical errors which need to be corrected. Overall, the clarity of the text is good and easily understandable.

Author Response

Thank you for your time devoted to reviewing our manuscript.

We are grateful for all your comments.

We have highlighted changes made to the main text in the red color.

  1. Figure 1 / The image is not clear to understand. Please mark the pale optic disk and the marking of the advanced glaucomatous neuropathy.

We have made modifications to Figure 1 and its description.

  1. Please include a supplementary table with all TREX1 gene mutations previously reported for AGS. 

Thank you very much for this suggestion. This table has been added.

  1. Page 4/ Line 137 / “The child developed exposure keratopathy OU due to slight lagophthalmos during sleep”  - Could you please explain what are you trying to establish with this statement?

Thank you for your remark. We introduced the change in the main text to make the sentence more understandable (line 139)

The manuscript has very few typographical and grammatical errors which need to be corrected. Overall, the clarity of the text is good and easily understandable.

Language proofreading of the text was conducted.